# Synthesis of Novel *α*-Trifluoroanisole Derivatives Containing Phenylpyridine Moieties with Herbicidal Activity

**DOI:** 10.3390/ijms231911083

**Published:** 2022-09-21

**Authors:** Zengfei Cai, Yangyang Cao, Xiaohua Du

**Affiliations:** Catalytic Hydrogenation Research Center, Zhejiang Key Laboratory of Green Pesticides and Cleaner Production Technology, Zhejiang Green Pesticide Collaborative Innovation Center, Zhejiang University of Technology, Hangzhou 310014, China

**Keywords:** *α*-trifluoroanisole, phenylpyridine, synthesis, herbicidal activity

## Abstract

To find novel herbicidal compounds with high activity and broad spectrum, a series of phenylpyridine moiety-containing *α*-trifluoroanisole derivatives were designed, synthesized, and identified via nuclear magnetic resonance (NMR) and high-resolution mass spectrometry (HRMS). Greenhouse-based herbicidal activity assays revealed that compound **7a** exhibited > 80% inhibitory activity against *Abutilon theophrasti*, *Amaranthus retroflexus*, *Eclipta prostrate*, *Digitaria sanguinalis*, and *Setaria viridis* at a dose of 37.5 g a.i./hm^2^, which was better than fomesafen. Compound **7a** further exhibited excellent herbicidal activity against *Abutilon theophrasti* and *Amaranthus retroflexus* in this greenhouse setting, with respective median effective dose (ED_50_) values of 13.32 and 5.48 g a.i./hm^2^, both of which were slightly superior to fomesafen (ED_50_ = 36.39, 10.09 g a.i./hm^2^). The respective half-maximal inhibitory concentration (IC_50_) for compound **7a** and fomesafen when used to inhibit the *Nicotiana tabacum* protoporphyrinogen oxidase (*Nt*PPO) enzyme, were 9.4 and 110.5 nM. The docking result of compound **7a** indicated that the introduction of 3-chloro-5-trifluoromethylpyridine and the trifluoromethoxy group was beneficial to the formation of stable interactions between these compounds and *Nt*PPO. This work demonstrated that compound **7a** could be further optimized as a PPO herbicide candidate to control various weeds.

## 1. Introduction

Species regarded as weeds generally exhibit advantages with respect to their ability to compete for water, space, light, and nutrients as compared to crops thereby resulting in crop production losses [1,2]. The Food and Agriculture Organization of the United Nations (FAO) estimates that weeds, pests, and plant diseases result in 20–40% reductions in annual global crop yields [3]. Herbicides can effectively suppress weed growth, thereby improving crop yields. FAO data suggest that sustained herbicide use would improve global food production by a minimum of 10% per year [4]. Therefore, it is necessary to continuously develop new herbicides with high efficiency and a broad spectrum to deal with various weeds.

The presence of the trifluoromethoxy moiety yields *α*-trifluoroanisole compound properties, including electron-withdrawing [5], lipophilicity [6], and metabolic stability. Several pesticides containing *α*-trifluoroanisole structures have been successfully synthesized to date [7,8,9,10,11,12,13,14], including indoxacarb, triflumuron, metaflumizone, flufenerim, flometoquin, flucarbazone-sodium, thifluzamide, and flurprimidol (Figure 1).

Protoporphyrinogen oxidase (PPO) is the last shared enzyme in the plant chlorophyll and animal heme biosynthesis pathways, which catalyzes the oxidation of protoporphyrinogen IX to protoporphyrin IX. Inhibiting the activity of PPO will lead to the massive accumulation of protoporphyrinogen IX, which penetrates the cytoplasm and is oxidized to protoporphyrin IX via other non-enzymatic mechanisms or oxidases. Protoporphyrin IX can generate reactive oxygen species under light, thereby resulting in the death of the plant. Therefore, PPO has been an important target for a variety of herbicides. Commercially available PPO inhibitors with herbicidal activity can be classified into oxadiazoles, uracils, phenylpyrazoles, diphenyl ethers, thiadiazoles, triazolinones, oxazolidinediones, and *N*-phenylphthalimides based upon their structural characteristics [15,16,17,18,19,20,21]. These herbicides have several advantages, including rapid onset of action, prolonged efficacy, a low environmental impact, and minimal mammalian toxicity [22]. Phenylpyridine compounds, which are structurally similar to uracil and diphenyl ethers, have been studied in the context of pesticide development. Schaefer et al. demonstrated that synthesized substituted 2-phenylpyridines exhibited excellent PPO enzyme inhibitory activity and herbicidal activity [22,23]. Substituted 3-(pyridin-2-yl)benzenesulfonamide derivatives prepared by Liu et al. exhibited effective inhibition against various weeds, especially broadleaf weeds [24,25,26]. In addition, Matsuya et al. demonstrated that *N*-[4-(arylmethyloxy)phenyl]-4-methyl-3,4,5,6-tetrahydrophthalimide exhibited good herbicidal activity [27,28]. Theodoridis reported that 2-[(4-heterocyclic-substituted-3-halophenoxymethyl)phenoxy]alkanoates showed excellent herbicidal activities against broadleaf weeds for both pre- and postemergence [29]. Therefore, *α*-trifluoroanisole was introduced into the active structure of 2-phenylpyridines to discover novel broad-spectrum herbicides.

Here, 18 novel compounds were synthesized via the introduction of substituted *α*-trifluoroanisole moieties into 2-phenylpyridines, and the resultant compounds were found to exhibit herbicidal activity against both broadleaf and grass weeds. The synthesis strategy for these compounds is detailed in Figure 2.

## 2. Results and Discussion

### 2.1. Synthesis of Target Compounds

The synthesis procedures for the target compounds used in this study are outlined in Figure 1. Intermediates **3a**–**3i** were synthesized via a Suzuki cross-coupling reaction from 2,3-dichloro-5-trifluoromethylpyridine **1** and substituted p-hydroxyphenylboronic acid **2a**–**2i**, while intermediates **6j**–**6r** were synthesized using substituted 4-trifluoromethoxybenzaldehyde as starting materials in a simple two-step reaction. These approaches yielded high levels of compounds **7a**–**7r** using intermediates **3a**–**3i** and intermediates **6a**, **6j**–**6r** as materials via nucleophilic substitution reactions in which *N*,*N*-Dimethylformamide DMF served as a solvent. After synthesis, compounds were characterized via HRMS and NMR, with the resultant spectral and analytical data being consistent with the expected structures. X-ray diffraction crystallography was further used to confirm the structure of compound **7a** (Figure 3). The crystal data for compound **7a** are shown in Appendix A, and the NMR and HRMS spectra of compounds **7a**–**7r** are shown in Appendix A.

### 2.2. Greenhouse Herbicidal Activity Assays

Herbicidal activity levels for compounds **7a**–**7r**, when used for the post-emergence treatment of dicotyledonous weeds *Abutilon theophrasti* (*A. theophrasti*), *Amaranthus retroflexus* (*A. retroflexus*), *Eclipta prostrate* (*E. prostrate*), and monocotyledonous weeds *Digitaria sanguinalis* (*D. sanguinalis*), *Echinochloa crusgalli* (*E. crusgalli*), *Setaria viridis* (*S. viridis*) were next assessed (Table 1), with fomesafen serving as a control. The majority of these target compounds exhibited excellent inhibitory activity against all tested dicotyledonous weeds, with some additionally exhibiting inhibitory activity against all tested monocotyledonous weeds. Of these, compounds **7a**, **7i**, and **7r** achieved 100% inhibitory activity when used for the post-emergence treatment of *A. retroflexus*, *A. theophrasti*, and *E. prostrata* at 150 g a.i./hm^2^, compounds **7b**, **7c**, **7j**, **7k**, **7l**, **7m**, **7n**, **7o**, **7p**, and **7q** also exhibited > 80% activity against these broadleaf weeds. In addition, compounds **7a** and **7j** also effectively suppressed the growth of all tested weeds, whereas compounds **7b**, **7c**, **7i**, **7k**, and **7n** exhibited > 80% inhibitory activity against all tested target weeds other than *E. crusgalli*. Other compounds exhibited varying levels of general herbicidal activities.

Further herbicidal activities revealed that at a 37.5 g a.i./hm^2^ dose, compounds **7a**, **7j**, and **7k** achieved 100% inhibition against the broadleaf weeds *A. theophrasti* and *A. retroflexus* in post-emergence testing, performing better than fomesafen, with compound **7a** additionally exhibiting > 80% inhibitory activities against the broadleaf weed *E. prostrata* and the grass weeds *D. sanguinalis* and *S. Viridis* (Table 2).

SAR analysis indicated that the activities of these compounds were dependent on R_1_, R_2_, R_3_, and R_4_. For R_1_ and R_2_ substitutions, optimal herbicidal activity was observed when both R_1_ and R_2_ were hydrogen or fluorine atoms. In contrast, such activity decreased markedly with R_2_ substitution of bulky groups, such as CF_3_ and CH_3_. With respect to the R_3_ and R_4_ substitutions, optimal herbicidal activity was observed when R_4_ was a hydrogen atom or chlorine atom, whereas other substitutions were associated with reduced activity against grass weeds. As such, the best herbicidal activity for the synthesized compounds was observed when both R_1_, R_2_, R_3_, and R_4_ were substituted with hydrogen atoms.

Compound **7a** was next used in weed spectrum control and crop injury tests conducted under greenhouse conditions (Figure 4). Compound **7a** exhibited good broad-spectrum inhibitory activity against the majority of tested broadleaf weeds and several grass weeds, with over 70% herbicidal activity at a dose of 18.75 g a.i./hm^2^. In addition, compound **7a** was able to fully control the growth of *Medicago sativa*, *Portulaca oleracea*, *Abutilon theophrasti*, *Amaranthus retroflexus*, *Cyperus rotundus L.*, and *Solanum nigrum*. Compound **7a** also exhibited slight damage to the tested crops, with 20% inhibitory effect on *Zea mays* (maize, *Z. mays*), *Triticum aestivum* (wheat, *T. aestivum*), *Gossypium* spp. (cotton, *G.* spp.), and 15% inhibitory effect on *Orysa sativa* (rise, *O. sativa*), *Glycine max* (soybean, *G. max*).

To evaluate the ability of compound 7a more fully, i.e., to inhibit the growth of *A. theophrasti* and *A. retroflexus*, an ED_50_ assay was next conducted (Table 3). Compound **7a** exhibited excellent herbicidal activities against *A. theophrasti* and *A. retroflexus,* with respective ED_50_ values of 13.32 and 5.48 g a.i./hm^2^, performing slightly better than the control fomesafen (ED_50_ = 36.39, 10.09 g a.i./hm^2^). 

### 2.3. Analysis of In Vitro PPO Inhibitory Activity

Compound **7a** exhibited significant inhibitory activity against the *Nt*PPO enzyme in vitro (IC_50_ = 9.4 nM), with this value obviously being better than that for fomesafen (IC_50_ = 110.5 nM) (Table 4).

### 2.4. Docking Analysis

To better understand the mechanism of action between the prepared compounds and PPO, molecular docking was used to predict the binding conformation of small molecule compounds and target enzymes at the active site. Specifically, compound **7a** and fomesafen were selected for docking studies to predict the binding mode of all target compounds to PPO. As shown in Figure 5B, the phenyl ring-containing trifluoromethyl group of fomesafen forms a π–π stacking interaction with the amino acid residue Phe392, the fluorine atom of the trifluoromethyl group forms a hydrophobic interaction with the amino acid residue Leu356, the phenyl ring-containing nitro group forms a π–π stacking interaction with the coenzyme FAD600, the nitro group forms a hydrogen bond with the coenzyme FAD600, and the sulfonyl group forms two hydrogen bonds with the amino acid residue Arg98. As shown in Figure 5A, the phenyl ring containing trifluoromethoxy group of compound **7a** forms a π–π stacking interaction with the amino acid residue Phe392 and a hydrophobic interaction with the amino acid residue Leu356, the pyridine ring forms a π–π stacking interaction with the coenzyme FAD600, the phenyl ring linked to pyridine forms a hydrophobic interaction with the amino acid residue Leu372, the fluorine atoms of the trifluoromethoxy group form a hydrophobic interaction with the amino acid residue Phe439 and Leu334, respectively. Furthermore, the difference in herbicidal activity of compound **7a** and fomesafen may be attributed to the different interactions.

## 3. Materials and Methods

### 3.1. Instrumentation

All reagents and other materials were obtained from Guangdong Guanghua Sci-tech Co., Ltd. (Guangdong, China), Bide Pharmatech Co., Ltd. (Shanghai, China), and Accela ChemBio Co., Ltd. (Shanghai, China) and used without additional purification unless otherwise noted. A B-545 melting point instrument (Buchi, Hangzhou, China) was used to quantify melting points, which were used without further correction. A Bruker AV-400 or AV-500 MHz spectrometer (Bruker, Hangzhou, China) was used to generate NMR spectra, with CDCl_3_ or DMSO-*d_6_* serving as solvents. An Agilent 6545 Q-TOF LCMS spectrometer (Agilent, Hangzhou, China) was used for mass spectrometry. A Bruker D8 Venture diffractometer (Bruker, Hangzhou, China) was utilized to collect crystallographic data.

### 3.2. Synthesis 

The synthesis approach for *α*-trifluoroanisole derivatives in this study is outlined in Figure 1. Among these, compounds **6a**, **6j**, and **6k** were obtained from commercial sources and used without additional purification.

#### 3.2.1. General Approach to the Synthesis of Compounds **3a**–**3i**

To prepare these compounds, 2,3-dichloro-5-trifluoromethylpyridine **1** (5 mmol), potassium carbonate (10 mmol), triphenylphosphorus (0.5 mmol), substituted p-hydroxybenzeneboronic acid **2a**–**2i** (5.5 mmol), and palladium(II) acetate (0.25 mmol) were mixed for 6 h with 5 mL of CH_3_OH and 10 mL of CH_3_CN at 50 °C under N_2_. Following the completion of this reaction as determined via TLC, the mixture was extracted thrice using ethyl acetate. Organic layers from these three extraction steps were then combined, rinsed with brine, and dried over MgSO_4_. Vacuum filtration was used to remove solid residues, and the solvent was removed. The remaining residue was then recrystallized with ethanol and water at 70 °C to obtain compounds **3a**–**3i** [30].

#### 3.2.2. General Approach to the Synthesis of Compounds **5l**–**5r**, **6l**–**6r**

Sodium borohydride (10 mmol) was slowly added to a solution of substituted p-trifluoromethoxy benzaldehyde (5 mmol) in methanol (20 mL) with constant magnetic stirring at −10 °C over a 10 min period, followed by further stirring for an 8 h period. Following the completion of this reaction as determined via TLC, the mixture was evaporated until dry at which time HCl (3 N) was added to the remaining residue to yield an acidic solution that was subsequently extracted thrice using ethyl acetate. The combined ethyl acetate layers were then rinsed using brine, after which the organic layer was evaporated to yield compounds **5l**–**5r** for subsequent use without additional purification [31].

Phosphorus tribromide (5.4 mmol) in CH_2_Cl_2_ (2 mL) was added in a dropwise manner to a substituted 4-trifluoromethoxybenzyl alcohol **5l**–**5r** (3 mmol) in CH_2_Cl_2_ (10 mL) solution at −5 °C with stirring for 5 h. Following the completion of this reaction as determined via TLC, the mixture was extracted thrice with CH_2_Cl_2_. The combined dichloromethane layers were then rinsed using NaHCO_3_ and brine, dried over MgSO_4_, and concentrated to yield the substituted 4-trifluoromethoxybenzyl bromide **6l**–**6r** for subsequent use without additional purification [31].

#### 3.2.3. General Approach to Synthesis of Compounds **7a**–**7r**

Substituted phenylpyridines 3a–3i (2 mmol), *N*,*N*-dimethylformamide (10 mL), and NaH (3 mmol, 0.12 g) were mixed and stirred in a three-necked reaction flask at 20 °C for 30 min under N_2_. Next, the substituted 4-trifluoromethoxybenzyl bromide 6a, 6j–6r (2.4 mmol) was added followed by stirring for 8 h at 60 °C. Following the completion of this reaction as determined via TLC, the mixture was extracted thrice using ethyl acetate (30 mL × 3). The pooled ethyl acetate layers were then rinsed thrice using brine and concentrated. Residues were then purified via silica gel column chromatography using ethyl acetate (EA) and petroleum ether (PE) (V_EA_:V_PE_=1:15) to yield compounds 7a–7r [30].

***3-Chloro-2-(4-((4-(trifluoromethoxy)benzyl)oxy)phenyl)-5-(trifluoromethyl)pyridine* (7a):** White solid, yield 79.3%, m.p. 59.1-60.5 °C, ^1^H NMR (500 MHz, DMSO-*d_6_*) *δ*: 8.99 (d, *J* = 1.1 Hz, 1H), 8.51 (s, 1H), 7.78 − 7.75 (m, 2H), 7.65 − 7.61 (m, 2H), 7.41 (d, *J* = 8.0 Hz, 2H), 7.19 − 7.16 (m, 2H), 5.24 (s, 2H). ^13^C NMR (126 MHz, DMSO-*d_6_*) *δ*: 159.27, 158.81, 147.92 (q, *J* = 1.4 Hz), 144.26 (q, *J* = 3.9 Hz), 136.26, 135.68 (q, *J* = 3.5 Hz), 131.01, 129.56, 129.30, 129.16, 124.39 (q, *J* = 33.04 Hz), 122.87 (q, *J* = 271.67 Hz), 120.99, 120.05(q, *J* = 256.86 Hz), 114.35, 68.42. HRMS (ESI): calculated for C_20_H_13_ClF_6_NO_2_ [M+H]^+^ 448.0534, found 448.0534.

***3-Chloro-2-(2-methyl-4-((4-(trifluoromethoxy)benzyl)oxy)phenyl)-5-(trifluoromethyl)pyridine* (7b):** White solid, yield 93.5%, m.p. 67.8-69.4 °C, ^1^H NMR (400 MHz, DMSO-*d_6_*) *δ*: 9.01 (s, 1H), 8.58 (s, 1H), 7.62 (d, *J* = 8.5 Hz, 2H), 7.41 (d, *J* = 8.3 Hz, 2H), 7.22 (d, *J* = 8.4 Hz, 1H), 7.03 (d, *J* = 2.0 Hz, 1H), 6.96 (dd, *J* = 8.5, 2.3 Hz, 1H), 5.19 (s, 2H), 2.07 (s, 3H). ^13^C NMR (101 MHz, DMSO-*d_6_*) *δ*: 161.04, 158.60, 147.93 (q, *J* = 1.8 Hz), 144.19 (q, *J* = 4.0 Hz), 137.32, 136.54, 135.03 (q, *J* = 3.4 Hz), 131.11, 130.25, 129.95, 129.66, 125.11 (q, *J* = 33.0 Hz), 122.94 (q, *J* = 274.0 Hz), 121.13, 120.12 (q, *J* = 256.9 Hz), 116.27, 111.91, 68.29, 19.27. HRMS (ESI): calculated for C_21_H_14_ClF_6_NO_2_Na [M+Na]^+^ 484.0509, found 484.0505.

***3-Chloro-2-(3-fluoro-4-((4-(trifluoromethoxy)benzyl)oxy)phenyl)-5-(trifluoromethyl)pyridine* (7c):** White solid, yield 71.6%, m.p. 48.5-51.8 °C, ^1^H NMR (500 MHz, CDCl_3_) *δ*: 8.74 (s, 1H), 7.94 (d, *J* = 1.3 Hz, 1H), 7.56 (dd, *J* = 12.0, 2.0 Hz, 1H), 7.49 (d, *J* = 8.6 Hz, 1H), 7.42 (d, *J* = 8.5 Hz, 2H), 7.18 (s, 1H), 7.17 (s, 1H), 7.01 (t, *J* = 8.4 Hz, 1H), 5.13 (s, 2H). ^13^C NMR (126 MHz, CDCl_3_) *δ*: 158.07, 152.35 (d, *J* = 247.4 Hz), 149.30 (q, *J* = 1.7 Hz), 147.96 (d, *J* = 10.8 Hz), 144.40 (q, *J* = 3.8 Hz), 135.70 (q, *J* = 3.5 Hz), 134.99, 130.66 (d, *J* = 6.7 Hz), 130.06, 129.02, 126.20 (q, *J* = 33.8 Hz), 126.13 (d, *J* = 3.6 Hz), 122.84 (q, *J* = 273.3 Hz), 121.34, 120.63 (q, *J* = 257.8 Hz), 118.04 (d, *J* = 20.4 Hz), 115.02 (d, *J* = 1.6 Hz), 70.63. HRMS (ESI): calculated for C_20_H_11_ClF_7_NO_2_Na [M+Na]^+^ 488.0259, found 488.0255.

***3-Chloro-2-(3-chloro-4-((4-(trifluoromethoxy)benzyl)oxy)phenyl)-5-(trifluoromethyl)pyridine* (7d):** White solid, yield 99.1%, m.p. 60.2-62.0 °C, ^1^H NMR (400 MHz, DMSO-*d_6_*) *δ*: 9.01 (s, 1H), 8.56 (s, 1H), 7.86 (d, *J* = 1.9 Hz, 1H), 7.75 (dd, *J* = 8.6, 1.9 Hz, 1H), 7.64 (d, *J* = 8.5 Hz, 2H), 7.43 (d, *J* = 8.2 Hz, 2H), 7.39 (d, *J* = 8.7 Hz, 1H), 5.34 (s, 2H). ^13^C NMR (101 MHz, DMSO-*d_6_*) *δ*: 157.56, 154.48, 148.04 (q, *J* = 1.4 Hz), 144.45 (q, *J* = 3.9 Hz), 135.97 (q, *J* = 3.4 Hz), 135.85, 130.98, 130.20, 129.75, 129.50, 129.48, 124.91 (q, *J* = 33.1 Hz), 122.87 (q, *J* = 274.0 Hz), 121.23, 120.11 (q, *J* = 257.3 Hz), 113.63, 69.29. HRMS (ESI): calculated for C_20_H_12_Cl2F_6_NO_2_ [M+H]^+^ 482.0144, found 482.0144.

***3-Chloro-2-(3-nitro-4-((4-(trifluoromethoxy)benzyl)oxy)phenyl)-5-(trifluoromethyl)pyridine (7e):*** White solid, yield 71.8%, m.p. 122.1-124.8 °C, ^1^H NMR (400 MHz, DMSO-*d_6_*) *δ*: 9.06 (s, 1H), 8.63 (s, 1H), 8.34 (d, *J* = 2.2 Hz, 1H), 8.11 (dd, *J* = 8.8, 2.2 Hz, 1H), 7.63 (m, 3H), 7.45 (d, *J* = 8.4 Hz, 2H), 5.45 (s, 2H). ^13^C NMR (101 MHz, DMSO-*d_6_*) *δ*: 156.78, 151.74, 148.12 (q, *J* = 1.2 Hz), 144.63 (q, *J* = 4.0 Hz), 139.04, 136.12 (q, *J* = 3.5 Hz), 135.44, 135.31, 129.73, 129.46, 129.20, 126.36, 125.32 (q, *J* = 33.1 Hz), 122.84 (q, *J* = 274.2 Hz), 121.29, 120.11 (q, *J* = 257.5 Hz), 115.28, 69.86. HRMS (ESI): calculated for C_20_H_12_ClF_6_N_2_O_4_ [M+H]^+^ 493.0384, found 493.0378.

***3-Chloro-2-(4-((4-(trifluoromethoxy)benzyl)oxy)-3-(trifluoromethyl)phenyl)-5-(trifluoromethyl)pyridine* (7f):** White solid, yield 40.8%, m.p. 73.3-75.1 °C, ^1^H NMR (400 MHz, DMSO-*d_6_*) *δ*: 9.02 (s, 1H), 8.58 (s, 1H), 8.08 (d, *J* = 8.7 Hz, 1H), 8.04 (s, 1H), 7.60 (d, *J* = 8.5 Hz, 2H), 7.51 (d, *J* = 8.8 Hz, 1H), 7.43 (d, *J* = 8.2 Hz, 2H), 5.41 (s, 2H). ^13^C NMR (101 MHz, DMSO-*d_6_*) *δ*: 157.47, 156.75, 148.02 (q, *J* = 1.4 Hz), 144.54 (q, *J* = 4.0 Hz), 136.01 (q, *J* = 3.4 Hz), 135.66, 135.41, 129.56, 129.13, 129.01, 128.20 (q, *J* = 5.3 Hz), 125.05 (q, *J* = 33.2 Hz), 123.50 (q, *J* = 273.6 Hz), 122.86 (q, *J* = 274.0 Hz), 121.24, 120.11 (q, *J* = 257.2 Hz), 117.06 (q, *J* = 30.6 Hz), 113.79, 69.20. HRMS (ESI): calculated for C_21_H_11_Cl_2_F_9_NO_2_ [M+Cl]^−^ 550.0029, found 550.0036.

***3-Chloro-2-(3-methyl-4-((4-(trifluoromethoxy)benzyl)oxy)phenyl)-5-(trifluoromethyl)pyridine* (7g):** White solid, yield 54.3%, m.p. 58.8-61.2 °C, ^1^H NMR (500 MHz, CDCl_3_) *δ*: 8.81 (d, *J* = 0.9 Hz, 1H), 8.00 (d, *J* = 1.6 Hz, 1H), 7.66 − 7.60 (m, 2H), 7.49 (d, *J* = 8.6 Hz, 2H), 7.26 (s, 1H), 7.24 (s, 1H), 6.96 (d, *J* = 9.1 Hz, 1H), 5.15 (s, 2H), 2.36 (s, 3H). ^13^C NMR (126 MHz, CDCl_3_) *δ*: 159.59, 157.92, 148.92 (q, *J* = 1.7 Hz), 144.13 (q, *J* = 3.9 Hz), 135.76, 135.34 (q, *J* = 3.4 Hz), 132.10, 129.91, 129.50, 128.53, 128.51, 127.14, 125.51 (q, *J* = 33.7 Hz), 122.87 (q, *J* = 273.2 Hz), 121.15, 120.54 (q, *J* = 257.7 Hz), 110.73, 69.13, 16.48. HRMS (ESI): calculated for C_21_H_15_ClF_6_NO_2_ [M+H]^+^ 462.0690, found 462.0690.

***5-(3-Chloro-5-(trifluoromethyl)pyridin-2-yl)-2-((4-(trifluoromethoxy)benzyl)oxy)benzonitrile* (7h):** White solid, yield 45.2%, m.p. 130.0-132.5 °C, ^1^H NMR (400 MHz, DMSO-*d_6_*) *δ*: 9.03 (s, 1H), 8.61 (s, 1H), 8.16 (d, *J* = 2.1 Hz, 1H), 8.08 (dd, *J* = 8.9, 2.2 Hz, 1H), 7.65 (d, *J* = 8.5 Hz, 2H), 7.51 (d, *J* = 9.0 Hz, 1H), 7.45 (d, *J* = 8.2 Hz, 2H), 5.42 (s, 2H). ^13^C NMR (101 MHz, DMSO-*d_6_*) *δ*: 160.49, 157.11, 148.18 (q, *J* = 1.4 Hz), 144.55 (q, *J* = 4.0 Hz), 136.25, 136.00 (q, *J* = 3.4 Hz), 135.31, 134.80, 129.91, 129.69, 129.64, 125.23 (q, *J* = 33.2 Hz), 122.84 (q, *J* = 273.8 Hz), 121.29, 120.10 (q, *J* = 257.5 Hz), 115.83, 113.37, 100.76, 69.58. HRMS (ESI): calculated for C_21_H_12_ClF_6_N_2_O_2_ [M+H]^+^ 473.0487, found 473.0486.

***3-Chloro-2-(2,3-difluoro-4-((4-(trifluoromethoxy)benzyl)oxy)phenyl)-5-(trifluoromethyl)pyridine* (7i):** White solid, yield 49.7%, m.p. 62.0-64.0 °C, ^1^H NMR (400 MHz, DMSO-*d_6_*) *δ*: 9.07 (s, 1H), 8.66 (s, 1H), 7.65 (d, *J* = 8.5 Hz, 2H), 7.43 (d, *J* = 8.1 Hz, 2H), 7.38 − 7.28 (m, 2H), 5.35 (s, 2H). ^13^C NMR (101 MHz, DMSO-*d_6_*) *δ*: 154.70, 148.71 (dd, *J* = 7.6, 3.2 Hz), 148.23 (q, *J* = 1.5 Hz), 148.02 (dd, *J* = 249.4, 11.2 Hz), 144.71 (q, *J* = 3.6 Hz), 140.21 (dd, *J* = 247.2, 14.3 Hz), 135.46, 135.44 (q, *J* = 3.5 Hz), 131.44, 129.99, 125.96 (q, *J* = 33.3 Hz), 125.50 (t, *J* = 3.6 Hz), 122.76 (q, *J* = 274.4 Hz), 121.25, 120.10 (q, *J* = 256.8 Hz), 118.90 (d, *J* = 12.8 Hz), 110.60, 69.93. HRMS (ESI): calculated for C_20_H_10_Cl_2_F_8_NO_2_ [M+Cl]^−^ 517.9966, found 517.9976.

***3-Chloro-2-(4-((3-chloro-4-(trifluoromethoxy)benzyl)oxy)phenyl)-5-(trifluoromethyl)pyridine* (7j):** White solid, yield 91.5%, m.p. 62.3-63.8 °C, ^1^H NMR (400 MHz, DMSO-*d_6_*) *δ*: 8.98 (s, 1H), 8.52 (s, 1H), 7.81 (s, 1H), 7.75 (d, *J* = 8.5 Hz, 2H), 7.63 − 7.56 (m, 2H), 7.17 (d, *J* = 8.5 Hz, 2H), 5.24 (s, 2H). ^13^C NMR (101 MHz, DMSO-*d_6_*) *δ*: 159.09, 158.84, 144.37 (q, *J* = 4.0 Hz), 143.50 (q, *J* = 1.2 Hz), 138.40, 135.82 (q, *J* = 3.4 Hz), 131.13, 129.97, 129.50, 129.26, 128.13, 126.10, 124.46 (q, *J* = 33.4 Hz), 123.29, 122.94 (q, *J* = 274.2 Hz), 120.08 (q, *J* = 259.3 Hz), 114.40, 67.73. HRMS (ESI): calculated for C_20_H_12_Cl_2_F_6_NO_2_ [M+H]^+^ 482.0144, found 482.0145.

***3-Chloro-2-(4-((2-methoxy-4-(trifluoromethoxy)benzyl)oxy)phenyl)-5-(trifluoromethyl)pyridine* (7k):** White solid, yield 92%, m.p. 99.6-101.4 °C, ^1^H NMR (500 MHz, DMSO-*d_6_*) *δ*: 9.00 − 8.99 (m, 1H), 8.53 − 8.50 (m, 1H), 7.78 − 7.75 (m, 2H), 7.56 (d, *J* = 8.3 Hz, 1H), 7.17 − 7.14 (m, 2H), 7.08 (d, *J* = 1.9 Hz, 1H), 7.00 − 6.97 (m, 1H), 5.15 (s, 2H), 3.89 (s, 3H). ^13^C NMR (126 MHz, DMSO-*d_6_*) *δ*: 159.44, 158.82, 158.08, 149.18 (q, *J* = 1.4 Hz), 144.26 (q, *J* = 3.9 Hz), 135.70 (q, *J* = 3.4 Hz), 131.02, 130.24, 129.20, 129.14, 124.36 (q, *J* = 33.1 Hz), 122.87 (q, *J* = 273.4 Hz), 123.82, 120.05 (q, *J* = 256.9 Hz), 114.21, 112.09, 104.69, 64.19, 56.07. HRMS (ESI): calculated for C_21_H_15_ClF_6_NO_3_ [M+H]^+^ 478.0639, found 478.0639.

***3-Chloro-2-(4-((2-methyl-4-(trifluoromethoxy)benzyl)oxy)phenyl)-5-(trifluoromethyl)pyridine (******7l):*** White solid, yield 65%, m.p. 77.6-78.6 °C, ^1^H NMR (500 MHz, DMSO-*d_6_*) *δ*: 9.02 − 8.98 (m, 1H), 8.52 (d, *J* = 1.5 Hz, 1H), 7.79 − 7.76 (m, 2H), 7.58 (d, *J* = 8.4 Hz, 1H), 7.27 (s, 1H), 7.23 − 7.19 (m, 3H), 5.21 (s, 2H), 2.40 (s, 3H). ^13^C NMR (126 MHz, DMSO-*d_6_*) *δ*: 159.37, 158.82, 147.96 (q, *J* = 1.5 Hz), 144.26 (q, *J* = 3.8 Hz), 139.44, 135.69 (q, *J* = 3.5 Hz), 134.21, 131.00, 130.08, 129.32, 129.16, 124.37 (q, *J* = 33.1 Hz), 122.87 (q, *J* = 273.3 Hz), 122.34, 120.05 (q, *J* = 256.6 Hz), 117.98, 114.32, 67.20, 18.29. HRMS (ESI): calculated for C_21_H_15_ClF_6_NO_2_ [M+H]^+^ 462.0690, found 462.0693. 

***3-Chloro-2-(4-((3-fluoro-4-(trifluoromethoxy)benzyl)oxy)phenyl)-5-(trifluoromethyl)pyridine (*****7m*****):*** White solid, yield 57.4%, m.p. 73.9-75.1 °C, ^1^H NMR (400 MHz, DMSO-*d_6_*) *δ*: 8.99 (s, 1H), 8.53 (s, 1H), 7.75 (d, *J* = 8.7 Hz, 2H), 7.66 − 7.58 (m, 2H), 7.44 (d, *J* = 8.4 Hz, 1H), 7.17 (d, *J* = 8.8 Hz, 2H), 5.24 (s, 2H). ^13^C NMR (101 MHz, DMSO-*d_6_*) *δ*: 159.09, 158.86, 153.58 (d, *J* = 251.1 Hz), 144.39 (q, *J* = 4.0 Hz), 139.06 (d, *J* = 6.9 Hz), 135.84 (q, *J* = 3.4 Hz), 134.66 (dq, *J* = 11.5, 1.7 Hz), 131.13, 129.51, 129.27, 124.57 (d, J = 3.5 Hz), 124.39 (q, *J* = 33.5 Hz), 124.22, 122.95 (q, *J* = 274.2 Hz), 120.09 (q, *J* = 259.0 Hz), 116.60 (d, *J* = 19.1 Hz), 114.42, 67.88. HRMS (ESI): calculated for C_20_H_12_ClF_7_NO_2_ [M+H]^+^ 466.0439, found 466.0439.

***3-Chloro-2-(4-((2-fluoro-4-(trifluoromethoxy)benzyl)oxy)phenyl)-5-(trifluoromethyl)pyridine******(*****7n*****):*** White solid, yield 86.0%, m.p. 74.5-76.7 °C, ^1^H NMR (400 MHz, DMSO-*d_6_*) *δ*: 9.00 (s, 1H), 8.54 (s, 1H), 7.77 − 7.73 (m *J* = 3H), 7.48 (d, *J* = 10.4 Hz, 1H), 7.31 (d, *J* = 8.5 Hz, 1H), 7.19 (d, *J* = 8.8 Hz, 2H), 5.25 (s, 2H). ^13^C NMR (101 MHz, DMSO-*d_6_*) *δ*: 161.65, 159.15, 158.88, 148.80 (dq, *J* = 11.6, 1.3 Hz), 144.42 (q, *J* = 3.8 Hz), 135.87 (q, *J* = 3.4 Hz), 132.03, 131.16, 130.83 (d, *J* = 254.2 Hz), 129.30, 124.49 (q, *J* = 33.5 Hz), 123.34 (d, *J* = 14.7 Hz), 122.97 (q, *J* = 274.0 Hz), 119.96 (q, *J* = 258.5 Hz), 117.22 (d, *J* = 3.3 Hz), 114.34, 109.51 (d, *J* = 25.5 Hz), 63.16. HRMS (ESI): calculated for C_20_H_12_ClF_7_NO_2_ [M+H]^+^ 466.0439, found 466.0439. 

***2-(4-((3-Bromo-4-(trifluoromethoxy)benzyl)oxy)phenyl)-3-chloro-5-(trifluoromethyl)pyridine******(*****7o*****):*** White solid, yield 64.5%, m.p. 79.0-80.8 °C, ^1^H NMR (400 MHz, DMSO-*d_6_*) *δ*: 9.03 (s, 1H), 8.57 (s, 1H), 7.99 (s, 1H), 7.79 (d, *J* = 8.4 Hz, 2H), 7.64 (q, *J* = 8.3 Hz, 2H), 7.21 (d, *J* = 8.4 Hz, 2H), 5.28 (s, 2H). ^13^C NMR (101 MHz, DMSO-*d_6_*) *δ*: 159.11, 158.85, 144.88 (q, *J* = 1.3 Hz), 144.39 (q, *J* = 4.1 Hz), 138.53, 135.84 (q, *J* = 3.4 Hz), 133.03, 131.15, 129.50, 129.27, 128.80, 124.47 (q, *J* = 33.0 Hz), 122.95 (q, *J* = 273.7 Hz), 122.94, 120.07 (q, *J* = 259.3 Hz), 115.39, 114.41, 67.65. HRMS (ESI): calculated for C_20_H_12_BrClF_6_NO_2_ [M+H]^+^ 525.9639, found 525.9639. 

***3-Chloro-2-(4-((3-iodo-4-(trifluoromethoxy)benzyl)oxy)phenyl)-5-(trifluoromethyl)pyridine******(*****7p*****):*** White solid, yield 61.1%, m.p. 75.8-77.5 °C, ^1^H NMR (500 MHz, DMSO-*d_6_*) *δ*: 9.00 (s, 1H), 8.52 (s, 1H), 8.11 (d, *J* = 1.5 Hz, 1H), 7.77 (d, *J* = 8.7 Hz, 2H), 7.63 (dd, *J* = 8.4, 1.6 Hz, 1H), 7.48 (d, *J* = 8.3 Hz, 1H), 7.18 (d, *J* = 8.7 Hz, 2H), 5.22 (s, 2H). ^13^C NMR (126 MHz, DMSO-*d_6_*) *δ*: 159.10, 158.80, 148.12 (q, *J* = 1.5 Hz), 144.28 (q, *J* = 3.8 Hz), 138.98, 138.21, 135.72 (q, *J* = 3.3 Hz), 131.03, 129.42, 129.18, 124.40 (q, *J* = 33.1 Hz), 122.87 (q, *J* = 273.6 Hz), 121.41, 120.07 (q, *J* = 258.7 Hz), 114.36, 90.77, 67.52. HRMS (ESI): calculated for C_20_H_12_ClF_6_INO_2_ [M+H]^+^ 573.9500, found 573.9500. 

***3-Chloro-2-(4-((3-cyclopropyl-4-(trifluoromethoxy)benzyl)oxy)phenyl)-5-(trifluoromethyl)pyridine******(*****7q*****):*** White solid, yield 47.2%, m.p. 99.8-102.6 °C, ^1^H NMR (400 MHz, DMSO-*d_6_*) *δ*: 9.04 (s, 1H), 8.58 (s, 1H), 7.80 (d, *J* = 8.7 Hz, 2H), 7.45 − 7.37 (m, 2H), 7.21 (s, 2H), 7.19 (s, 1H), 5.21 (s, 2H), 2.18 − 2.09 (m, 1H), 1.10 − 1.04 (m, 2H), 0.79 (q, *J* = 5.1 Hz, 2H). ^13^C NMR (101 MHz, DMSO-*d_6_*) *δ*: 159.34, 158.90, 146.93 (q, *J* = 1.3 Hz), 144.39 (q, *J* = 4.0 Hz), 136.47, 136.43, 135.84 (q, *J* = 3.4 Hz), 131.09, 129.29, 129.25, 126.23, 125.17, 124.43 (q, *J* = 33.1 Hz), 122.96 (q, *J* = 273.7 Hz), 121.17, 120.41 (q, *J* = 257.1 Hz), 114.38, 68.63, 9.25, 8.62. HRMS (ESI): calculated for C_23_H_17_ClF_6_NO_2_ [M+H]^+^ 488.0847, found 488.0847. 

***3-Chloro-2-(4-((3-nitro-4-(trifluoromethoxy)benzyl)oxy)phenyl)-5-(trifluoromethyl)pyridine******(*****7r*****):*** White solid, yield 20.3%, m.p. 86.6-88.1 °C, ^1^H NMR (500 MHz, DMSO-*d_6_*) *δ*: 9.00 (d, *J* = 1.1 Hz, 1H), 8.54 (s, 1H), 8.33 (d, *J* = 2.1 Hz, 1H), 8.00 (dd, *J* = 8.5, 2.1 Hz, 1H), 7.81 (dd, *J* = 8.5, 1.1 Hz, 1H), 7.80 − 7.76 (m, 2H), 7.23 − 7.19 (m, 2H), 5.36 (s, 2H). ^13^C NMR (126 MHz, DMSO-*d_6_*) *δ*: 158.95, 158.82, 144.37 (q, *J* = 3.7 Hz), 142.14, 139.12 (q, *J* = 1.8 Hz), 138.37, 135.81 (q, *J* = 3.4 Hz), 134.12, 131.17, 129.66, 129.27, 124.99, 124.50 (q, *J* = 33.1 Hz), 123.86, 122.93 (q, *J* = 273.3 Hz), 119.82 (q, *J* = 259.8 Hz), 114.44, 67.44. HRMS (ESI): calculated for C_20_H_12_ClF_6_N_2_O_4_ [M+H]^+^ 493.0384, found 493.0383. 

Note: Due to the existence of the symmetrical structure and the multiplet caused by the fluorine atom, the number of signals in the NMR carbon spectrum of all the above compounds does not correspond to the number of carbons in the molecule.

### 3.3. Herbicidal Activities Assay

#### 3.3.1. Herbicidal Activities Glasshouse Assay

All weeds were obtained from the Pesticide Creation Center of the Zhejiang Research Institute of Chemical Industry. Levels of herbicidal activity for compounds 7a–7r against the monocotyledonous weeds *Digitaria sanguinalis* (*D. sanguinalis*), *Echinochloa crusgalli* (*E. crusgalli*), and *Setaria viridis* (*S. viridis*), and the dicotyledonous weeds *Abutilon theophrasti* (*A. theophrasti*), *Amaranthus retroflexus* (*A. retroflexus*), and *Eclipta prostrate* (*E. prostrate*) were assessed as in prior reports [30,32,33,34].

DMF was used to dissolve all test compounds, followed by their dilution with distilled water containing 0.1% Tween-80 to an appropriate dose. The application rates for post-emergence treatment were 150, 75, and 37.5 g a.i./ha. Seedlings were treated with test compounds at the three-leaf stage with three replicates, treated without test compound seedlings serving as blank controls, and fomesafen-treated seedlings serving as positive controls. The herbicidal activity was assessed after a 20-day period, with the results being compiled in Table 1 and Table 2. 

Growth inhibition rates (0–100%) were assessed based on/evaluated by a visual inspection, with 0% and 100%, respectively, corresponding to no damage and the total destruction of aboveground plant structures. 

#### 3.3.2. Weed Spectrum and Crop Injury Tests

Weed spectrum and crop injury tests were conducted for compound **7a** using the same procedures as above in a greenhouse, with an estimated post-emergence application rate of 18.75 g a.i./hm^2^. The utilized species included dicotyledonous weeds *Pharbitis nil* (*P. nil*), *Aeschynomene indica* (*A. indica*), *Veronica didyma* (*V. didyma*), *Eclipta prostrate* (*E. prostrate*), *Orychophragmus violaceus* (*O. violaceus*), *Chenopodium serotinum* (*C. serotinum*), *Galium spurium* L. (*G. spurium* L.), *Medicago sativa* (*M. sativa*), *Portulaca oleracea* (*P. oleracea*), *Abutilon theophrasti* (*A. theophrasti*), *Amaranthus retroflexus* (*A. retroflexus*), *Solanum nigrum* (*S. nigrum*); monocotyledonous weeds *Polypogon fugax* (*P. fugax*), *Setaria viridis* (*S. viridis*), *Eleusine indica* (*E. indica*), *Digitaria sanguinalis* (*D. sanguinalis*), *Pseudosorghum zollingeri* (*P. zollingeri*), *Poa annua* (*P. annua*), *Alopecurus aequalis* (*A. aequalis*), *Leptochloa panacea* (*L. panacea*), *Lolium perenne* (*L. perenne*), *Echinochloa crusgalli* (*E. crusgalli*); Cyperaceae weeds *Cyperus rotundus* L. (*C. rotundus* L.), and crop species *Orysa sativa* (rise, *O. sativa*), *Glycine max* (soybean, *G. max*), *Zea mays* (maize, *Z. mays*), *Triticum aestivum* (wheat, *T. aestivum*), *Gossypium* spp. (cotton, *G.* spp.). The results are summarized in Figure 4.

#### 3.3.3. Herbicidal Activities Glasshouse Assay against *Abutilon theophrasti* and *Amaranthus retroflexus*

To study the control effect of **7a** with the highest herbicidal activities against *Abutilon theophrasti* and *Amaranthus retroflexus*, its herbicidal activities were tested using the same procedures as above in a greenhouse, with fomesafen serving as a control. The application rates for post-emergence treatment were 450, 225, 150, 75, 45, 30, 15, 7.5, 3.75, and 1.875 g a.i./hm^2^. Seedlings were treated with test compounds at the three-leaf stage with three replicates, treated without test compound seedlings serving as blank controls, and fomesafen-treated seedlings serving as positive controls. The herbicidal activity was assessed after a 14-day period, with the results being compiled in Table 3. 

Then fresh weight inhibition rates were established by measuring fresh weight values for the aboveground parts of target weeds, and the ED_50_ values were calculated.

### 3.4. In Vitro Analysis of NtPPO Inhibitory Activity

To explore the mode of action more fully, i.e., for the prepared compounds, compound **7a** was selected as a representative target in effort to confirm its ability to inhibit PPO, with fomesafen serving as a control compound. *Nt*PPO was prepared as reported previously [35]. The inhibitory activities of the *Nt*PPO enzyme were determined using a previously reported method [36]. The *Nt*PPO catalytic rate was calculated by measuring changes in the concentrations of the *Nt*PPO catalytic product protoporphyrin IX over a given time period by measuring the absorbance at 410 nm via a UV–Vis spectrophotometer. 

### 3.5. Molecular Docking Analysis

The *Nt*PPO (PDB ID: 1SEZ) structure was obtained from the Protein Data Bank, with molecular docking analysis being performed using chain A of the *Nt*PPO structure [37]. The crystal structures of *Nt*PPO and the small molecule **7a** were obtained via a standard approach using Dock 6.9 (University of California, San Francisco, California 94158, USA) [38], with fomesafen serving as a control compound. The optimal binding conformation was then selected in light of the results of the docking score (Figure 5).

## 4. Conclusions

In conclusion, a series of new *α*-trifluoroanisole derivatives containing phenylpyridine moieties were herein developed and synthesized. Among these, compound **7a** exhibited optimal structural characteristics and good herbicidal activity when utilized for the post-emergence treatment of a wide spectrum of dicotyledonous weeds (at a dosage of 18.75 g a.i./hm^2^) in greenhouse testing. In addition, the excellent inhibitory activity against PPO and the docking result of compound **7a** will provide directions for further optimization as a candidate PPO herbicide. 

## Data Availability

Samples of the compounds are not available from the authors.

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
