# Peer review of "Synthesis of Novel α-Trifluoroanisole Derivatives Containing Phenylpyridine Moieties with Herbicidal Activity"

_ijms, 2022, doi:10.3390/ijms231911083_

Round 1
Reviewer 1 Report
This is a good quality paper that reports an extension of the authors' previous work on the study of the herbicidal activity of the synthesized compounds of various structures. The importance of these studies is beyond doubt, since the need for new and effective plant protection products in the world is very high. The results obtained are beyond doubt (synthesis, study of herbicidal activity, molecular docking). The article is logically structured.
I propose to accept this article in its current form, but I have a few comments:
1. In the ESI, I propose to add the name of the solvent in the headings of the spectra, since they are cut off for the convenience of the reader, but you have to constantly refer to the article in order to understand in which solvent the spectrum was recorded.
2. In Section 3, in the description of carbon spectra, the number of signals does not correspond to the number of carbons in the molecule. I know that the signals (quartets) that are formed due to the proximity to fluorine are difficult to accumulate, especially if they are quaternary. Perhaps, it is worth giving some explanations about this in the same section.
Reviewer 2 Report
The manuscript written by Cai et al. describes synthesis, herbicidal evaluation, and docking studies of a set of alpha-trifluoroanisole derivatives bearing phenylpyridine substituents. The topic is interesting and the reported results provide valuable information about antiherbal activity. Moreover, compound 7a exhibits commercial potential. The work fits within the scope of IJMS and I would support the paper for publication; however some issues need to be explained and/or corrected.
1. The reported molecules are potent herbicidal agents. How about their selectivity towards crops? Are they not toxic? In the further studies the authors could also focus on cytotoxicity, half-life time and the influence on the environment of the proposed compounds.
2. Table 3, equations y = ax + b – it is not clear nor how the parameters were calculated neither what is their meaning, since no description is provided. In this form, it gives no value added to the manuscript – please explain or erase them.
3. My internal suspiciously make me wonder why calculated and found HRMS data were in most cases EXACTLY the same? In addition, it is interesting that compounds 7a,j,k obtained from commercially available substrates were the most active. Is it possible that the purity of synthesized intermediates was not enough? Also, the NMR spectra provided in the Supporting Material are of low resolution – could the authors add zoom-in peaks?
To sum up, the manuscript requires revisions.
Author Response
请参阅附件。

Round 2
Reviewer 2 Report
The authors improved the manuscript. I see there is a statement in line 124 - "this dose resulted in slight crop injury". I suggest to broaden the discussion about results presented in Figure 4. Also, authors have added some zoom-in peaks to the HNMR spectra. The reported compounds are novel and innovative. I support the paper for publication in IJMS after minor revisions.
Author Response
We have broadened the discussion about results presented in Figure 4.